# *CD274* (*PD-L1*) Polymorphisms as Predictors of Efficacy in First-Line Platinum-Based Chemotherapy for Extensive-Stage Small Cell Lung Cancer

**DOI:** 10.3390/ijms26094245

**Published:** 2025-04-29

**Authors:** Andrés Barba, Laura López-Vilaró, Malena Ferre, Sergio Martinez-Recio, Margarita Majem, Ivana Sullivan, Juliana Salazar

**Affiliations:** 1Department of Medical Oncology, Hospital de la Santa Creu i Sant Pau, 08041 Barcelona, Spain; abarba@santpau.cat (A.B.); smartinezre@santpau.cat (S.M.-R.); mmajem@santpau.cat (M.M.); isullivan@santpau.cat (I.S.); 2Department of Medicine, Faculty of Medicine, Universitat Autònoma de Barcelona, 08035 Barcelona, Spain; 3Translational Medical Oncology Laboratory, Institut de Recerca Sant Pau (IR Sant Pau), 08041 Barcelona, Spain; 4Department of Pathology, Hospital de la Santa Creu i Sant Pau, 08041 Barcelona, Spain; llopezv@santpau.cat (L.L.-V.); mferref@santpau.cat (M.F.)

**Keywords:** extensive-stage small cell lung cancer, platinum-based chemotherapy, pharmacogenetics, biomarker, *CD274* gene, PD-L1

## Abstract

The cornerstone of first-line treatment in extensive-stage small cell lung cancer (ES-SCLC) is platinum- and etoposide-based chemotherapy. Platinum compounds could immunomodulate the tumor microenvironment in addition to their cytotoxic effect. Genetic variation in immune checkpoint (IC) pathways may predict chemotherapy efficacy. Polymorphisms in the IC genes were determined, and their association with survival was analyzed in 78 patients with ES-SCLC treated with chemotherapy. PD-L1 protein expression in tumor tissue was determined. Three variants in *CD274* were associated with better median progression-free survival (mPFS): rs2297136 (hazard ratio [HR] 0.52, 95% CI 0.29–0.93; *p* = 0.03), rs2282055 (HR 0.23, 95% CI 0.09–0.64; *p* = 0.005), and rs822336 (HR 0.41, 95% CI 0.23–0.73; *p* = 0.002). *CTLA4* rs231775 was also associated with mPFS (HR 0.30, 95% CI 0.14–0.63; *p* = 0.002). The variants *CD274* rs2297136 and *CD274* rs822336 were associated with platinum sensitivity (odds ratio [OR] 0.13, 95% CI 0.02–0.70; *p* = 0.02, and OR 0.08, 95% CI 0.01–0.46; *p* = 0.005, respectively). *CD274* rs2297136 was also associated with better overall survival (*p* = 0.02), but not after adjustment for covariates. No association was found between *CD274* germline variants and PD-L1 tumor expression. Our results suggest that *CD274* and *CTLA4* variants may be predictive biomarkers for platinum plus etoposide treatment in ES-SCLC.

## 1. Introduction

Small cell lung cancer (SCLC) accounts for 15% of all lung cancer cases [1] and is strongly related to tobacco exposure [2]. Platinum-based chemotherapy plus etoposide is the cornerstone of treatment for patients with untreated SCLC [3,4]. The addition of immune checkpoint (IC) inhibitors to chemotherapy has shown moderate improvement in progression-free survival (PFS) and overall survival (OS) in patients with extensive-stage (ES) disease [3,4,5]. These outcomes could be improved by introducing new drugs [6,7] or molecular biomarkers for accurate patient selection into current chemoimmunotherapy. Validation of new molecular classifications of SCLC tumors, such as those based on differential expression of transcriptional regulators [8,9], could also contribute to improved outcomes. Unfortunately, there are at present no predictors of efficacy for platinum-based treatments with or without immunotherapy in ES-SCLC. Programmed cell death ligand 1 (PD-L1) expression and tumor mutation burden status have been extensively investigated as immunotherapy biomarkers in some tumors [10,11]. Although PD-L1 expression has been validated in non-small cell lung cancer (NSCLC), neither of these two biomarkers has been confirmed in ES-SCLC and are therefore not recommended in therapeutic decision-making [12]. Other molecular markers have been postulated for platinum-based chemotherapy in NSCLC, such as excision repair cross-complementation group 1 (ERCC1) protein expression in tumor tissue [13] and *ERCC1* germline polymorphisms [14]. However, these markers have also not been validated in ES-SCLC [15].

In ES-SCLC, identifying and validating predictors for platinum-based chemotherapy is of great interest, as their implementation would enhance the synergistic clinical effect of chemotherapy and immunotherapy. This proposal is based on the knowledge that the mechanism of action of platinum compounds entails not only the formation of DNA adducts that induce cell cycle arrest and cell apoptosis in proliferating tumor cells [16], but also the modulation of the antitumor immune response [17,18,19]. The host immune response is tightly controlled by IC molecules both to prevent autoimmunity and to protect against infections and cancer. The programmed cell death protein-1 (PD-1) receptor, expressed on T cells, natural killer cells, monocytes, B cells, and dendritic cells, and its ligand PD-L1, expressed by antigen-presenting cells, are crucial in the negative regulation of the immune response [20]. However, in the tumor microenvironment, tumor cells also express these IC molecules as a mechanism to evade immune surveillance by inactivating cytotoxic T cells [21]. In this context, the efficacy of a platinum-based regimen may be conditioned by functional alterations of IC molecules [22].

Single-nucleotide polymorphisms (SNPs) in genes encoding ICs have been proposed to influence the efficacy of chemotherapy [23] and survival [24,25] in Asian patients with NSCLC and SCLC [26]. To the best of our knowledge, however, only the *CD155* rs1058402 variant has been associated with survival in ES-SCLC [26], although this variant has a minor allele frequency of less than 0.05 in European populations. Therefore, the rationale for this study is to explore new treatment-selection biomarkers in the IC pathways that may contribute to identifying a subset of ES-SCLC patients who would benefit most from platinum-containing regimens. Here, we examined a comprehensive set of SNPs in genes coding for receptors and ligands that mediate IC signaling in a retrospective cohort of ES-SCLC patients treated with first-line platinum-etoposide chemotherapy. In addition, we investigated whether there was a correlation between *CD274* (*PD-L1*) genetic variants and PD-L1 protein expression.

## 2. Results

### 2.1. Patients’ Characteristics and Clinical Variants Analyses

Table 1 shows the main characteristics of the ES-SCLC patients.

The median follow-up was 10.0 (interquartile range 6.9–15.8) months. The median OS was 9.7 months (95% confidence interval [CI] 7.8–11.5), and the median PFS was 6.1 months (95% CI 5.3–6.8). Patients with ECOG PS 0-1 had a better median OS than those with ECOG ≥ 2 (11.5 months [95% CI 6.9–16.2] versus 8.2 months [95% CI 7.6–8.7]; *p* = 0.001). No differences were observed in median PFS according to ECOG (*p* = 0.3). Patients treated with consolidative thoracic radiation therapy (TRT) had a better median OS (17.1 months [95% CI 7.0–27.1] versus 8.4 months [95% CI 6.6–10.3]; *p* = 0.002) and median PFS (7.3 months [95% CI 6.2–8.4] versus 5.2 months [95% CI 4.6–5.9]; *p* = 0.002) than patients who did not. No differences were found in median PFS or median OS according to sex (*p* = 0.7 and *p* = 0.4, respectively) or type of platinum (*p* = 0.08 and *p* = 0.07, respectively).

The analyses showed that 36.5% of patients were platinum-based chemotherapy sensitive. There was a significant positive association between platinum sensitivity and consolidative TRT (odds ratio [OR] 0.16, 95% CI 0.05–0.49; *p* < 0.001). Numerical differences were found between platinum sensitivity and cisplatin treatment (OR 0.37, 95% CI 0.14–1.00; *p* = 0.05).

### 2.2. Polymorphisms of Immune Checkpoint Genes and Progression-Free Survival Analyses

The full results of the univariate analyses between the genetic variants of IC signaling molecules and PFS are shown in Appendix A. *CD274* rs2282055 was the only variant that reached the Bonferroni-adjusted significance level for multiple comparisons (*p* < 0.001). Multivariate analyses for PFS showed significant associations for the rs2297136, rs2282055, and rs822336 variants in the *CD274* gene and the *CTLA4* rs231775 variant (Table 2).

For *CD274* rs2297136, patients with the AG and GG genotypes had a longer median PFS (6.3 months) than patients with the AA genotype (4.6 months) (hazard ratio [HR] 0.52, 95% CI 0.29–0.93; *p* = 0.03 in a dominant model) (Figure 1A). For *CD274* rs2282055, patients with TT and TG genotypes had a longer median PFS (6.1 months) than patients with the GG genotype (4.6 months) (HR 0.23, 95% CI 0.09–0.64; *p* = 0.005 in a recessive model) (Figure 1B). For *CD274* rs822336, patients with the GC and CC genotypes had a longer median PFS (6.5 months) than patients with the GG genotype (4.8 months) (HR 0.41, 95% CI 0.23–0.73; *p* = 0.002 in a dominant model) (Figure 1C). For *CTLA4* rs231775, patients with the AA and AG genotypes had a longer median PFS (6.1 months) than patients with the GG genotype (4.6 months) (HR 0.30, 95% CI 0.14–0.63; *p* = 0.002 in a recessive model) (Figure 1D).

### 2.3. Polymorphisms of Immune Checkpoint Genes and Overall Survival Analyses

Regarding OS, the full results of the univariate analyses are shown in Appendix A. Multivariate analyses showed marginal differences for *CD274* rs2297136: patients with the AG and GG genotypes had longer median OS (10.4 months) than patients with the AA genotype (7.5 months) (HR 0.62, 95% CI 0.36–1.09; *p* = 0.1 in a dominant model) (Appendix A).

### 2.4. Haplotype Analyses for CD274 Variants and Survival

Haplotype analyses were performed to determine the effect of *CD274* variants (rs2297136|rs2282055|rs822336) on PFS and OS. The GGG haplotype (frequency = 6%) was associated with median PFS (*p* = 0.009 univariate; *p* = 0.01 multivariate) and OS (*p* < 0.001 univariate; *p* < 0.001 multivariate) (Appendix A).

### 2.5. Polymorphisms of Immune Checkpoint Genes and Platinum Sensitivity Analyses

Analyses between platinum sensitivity and the four genetic variants associated with PFS showed significant associations for rs2297136 and rs822336 variants in the *CD274* gene when the covariates were included (Appendix A). Patients carrying the *CD274* rs2297136-G allele (OR 0.13, 95% CI 0.02–0.70; *p* = 0.02 in a dominant model) and those carrying the *CD274* rs822336-C allele (OR 0.08, 95% CI 0.01–0.46; *p* = 0.005 in a dominant model) were more likely to be platinum-sensitive.

### 2.6. Analyses of PD-L1 Protein Expression and CD274 Polymorphisms

Fifty-two tumor tissue samples (66.7%) were evaluable for PD-L1 expression analysis. The positivity of PD-L1 expression by the Tumor Proportion Score (TPS) was 3.8%, and no patients presented PD-L1 ≥ 50%. The positivity of PD-L1 expression by the Combined Positive Score (CPS) was 23.1%.

There was no correlation between *CD274* germline variants and PD-L1 tumor expression.

## 3. Discussion

In this retrospective cohort of ES-SCLC patients treated with first-line platinum-etoposide chemotherapy, some variants in the *CD274* (also known as *PD-L1*) and *CTLA4* genes were significantly associated with PFS.

ES-SCLC is an aggressive disease with a poor prognosis [27]. As the addition of IC inhibitors to platinum-based chemotherapy has provided a modest benefit, several treatment options are currently being investigated to improve survival in these patients [6,7]. In this scenario, platinum doublet remains the cornerstone of first-line treatment and a second-line choice for those patients with relapsed disease who are sensitive to platinum [28]. However, to date, no biomarkers have been identified to explain the interindividual variability in response to platinum compounds in ES-SCLC.

In lung cancer, few studies [23,24,25,26] have analyzed genetic variants in genes of the immune checkpoint signaling pathway based on the immunomodulatory effects of chemotherapy on the tumor microenvironment [17,18,19]. These studies have identified associations between chemotherapy response or survival and the *CD155* rs1058402 and *CD226* rs763361 variants in SCLC, although only the *CD155* rs1058402 variant was associated with OS in extensive-stage disease [26], and variants in the *CD274* gene in NSCLC [23,24,25]. In patients with advanced-stage NSCLC, the *CD274* rs2297136-G allele was found to be associated with chemotherapy response and OS [23]. In addition, in postoperative NSCLC patients treated with adjuvant platinum-based chemotherapy, patients carrying GC and CC genotypes for *CD274* rs822336 showed better survival [25].

Our results are consistent with those reported in NSCLC [23,25] and show that variants in the *CD274* gene may be feasible biomarkers of platinum-based chemotherapy efficacy in ES-SCLC. Patients carrying the rs2297136-G allele and patients carrying the rs822336-C allele had a longer median PFS, and they were more likely to respond to platinum-based chemotherapy. In addition, carriers of the rs2297136-G allele had a longer median OS, although this was not statistically significant, a result in line with the significant association observed with OS of the low-frequency haplotype GGG that includes the protective G allele for this variant. These findings suggest that the *CD274* variants might help identify patients with a higher probability of longer PFS and platinum-sensitive disease, thus lending sensitivity to second-line treatments, especially in those patients who received platinum rechallenge [29].

The *CD274* gene encodes the immune inhibitory ligand PD-L1, which binds to the PD-1 receptor, blocking T-cell activation to prevent autoimmunity, a mechanism also used by the tumor to escape immune surveillance [30]. *CD274* rs2282055 is an intron variant [31]. The prediction scores (rank 1f and score 0.55) of the variant from the Regulome database [32] show that there is evidence of its putative functionality. The *CD274* rs2297136 variant located in the 3′-UTR region may alter the microRNA binding sites for miR-324-5p and miR-296-5p and thus affect mRNA translational repression or destabilization [33]. In silico studies predicted a higher minimum free energy of hybridization of miR-324-5p and mRNA for the G allele [23], and in vitro studies showed that miR-296-5p suppressed the expression of the construct containing the G allele [34]. *CD274* rs822336 is an upstream variant, and the C allele may confer reduced PD-L1 mRNA expression [25] and promoter activity [24]. Thus, these germline *CD274* variants may be functional and could lead to decreased PD-L1 protein levels.

Based on these findings, PD-L1 in immune cells may therefore have an effect on the response to platinum compounds. Notably, in SCLC, PD-L1 expression occurs predominantly in immune cells rather than in tumor cells [35], so down-regulation of the PD-1/PD-L1 pathway may induce host immune responses and contribute to the effect of the antitumor activity of chemotherapy observed in our study. However, in the present study, no correlation was found between *CD274* germline variants and PD-L1 expression in a subgroup of patients. It should be noted that PD-L1 expression may be modified by therapeutic exposure to chemotherapy with or without immunotherapy or by disease progression. In the tumor microenvironment, transcription, post-transcription, and post-translation processes determine PD-L1 expression [36]. Thus, the low levels of PD-L1 protein expression and its possible temporal variation throughout disease progression may explain the lack of correlation observed in our study between germline *CD274* SNPs and protein expression at diagnosis.

Another result in this study was that patients with the AA and AG genotypes for the *CTLA4* rs231775 variant had longer median PFS than homozygous patients for the G allele. The *CTLA4* gene encodes for a protein that is expressed in activated T cells and is involved in attenuating the immune response. The *CTLA4* rs231775 A > G is a missense polymorphism (p.Thr17Ala) located in exon 1 [37]. However, in vitro experiments have shown that the protein expression is reduced in GG homozygotes compared to AA homozygotes [38], and further investigation is required before considering the variant a biomarker of platinum efficacy.

Platinum compounds have a cytotoxic effect on cancer cells but also act as host immune modulators. Cisplatin has been shown to increase major histocompatibility complex class I, enhance T-cell infiltration, proliferation, and cytotoxic activity, and decrease the expression of immunosuppressive cells, including myeloid-derived suppressor cells and Treg cells [17].

The findings of the present study are consistent with this knowledge and support the suggestion that the common genetic variation in the IC pathway would help identify those ES-SCLC patients treated with platinum compounds who have a favorable response. The study cohort was collected prior to the approval of combination chemotherapy with immunotherapy in ES-SCLC. This design allowed us to investigate predictive biomarkers for platinum-based chemotherapy without the bias that adding immunotherapy might have introduced, especially in a study focusing on the evaluation of genes in IC signaling pathways.

This pharmacogenetic study has some limitations. First, the sample size was modest, although the statistical power was sufficient for the objectives of the study. In addition, it was a cohort of patients from a single center and homogeneously treated. Second, the analyses did not show a correlation between germline *CD274* variants and PD-L1 expression in the tumor at diagnosis, probably due to low PD-L1 protein expression in the tumor, and further experiments would be necessary to elucidate the underlying mechanisms.

## 4. Materials and Methods

### 4.1. Study Population

This was a retrospective association study involving 78 patients with ES-SCLC treated between September 2010 and May 2021 at Hospital de la Santa Creu i Sant Pau (Barcelona, Spain). The study included patients treated with standard carboplatin/cisplatin-based chemotherapy plus etoposide for first-line treatment who had received at least two cycles of platinum-based chemotherapy and excluded those who received IC treatment. The inclusion of patients treated with consolidative TRT after first-line platinum-based chemotherapy was permitted. Patients previously treated with chemotherapy for limited-stage SCLC were excluded from the study.

### 4.2. Definitions of Clinical Endpoints

PFS was defined as the time from the date of initiation of chemotherapy to the date of disease progression, as defined by RECIST 1.1 [39], or death from any cause, whichever occurred first. OS was defined as the time from the date of initiation of chemotherapy to the date of death from any cause. Platinum sensitivity status was dichotomized into platinum-sensitive and platinum-resistant. Platinum-sensitive was defined as progression ≥ 90 days from the last dose of platinum-based chemotherapy, and platinum-resistant was defined as progression < 90 days from the last dose of platinum-based chemotherapy [40]. Patients who progressed during platinum-based chemotherapy (platinum-refractory) were classified as platinum-resistant patients.

### 4.3. Selection of Polymorphisms

Fifteen SNPs were selected in the *CD274* (*PD-L1*), *PDCD1*, *CTLA4*, *CD226*, and *LAG3* genes encoding IC signaling molecules. These are SNPs with functional evidence [41] or previously associated with susceptibility, prognosis, or treatment response in autoimmune diseases and some cancers (see Table 3).

### 4.4. Genotyping

DNA from whole blood samples was extracted automatically (Autopure LS system, Qiagen, Hilden, Germany). The genotyping of the samples was carried out by real-time PCR using TaqMan^®^ SNP genotyping assays on a 7900 HT Real Time PCR System (Applied Biosystems, Foster City, CA, USA) according to the manufacturers’ instructions. All the samples were successfully genotyped.

The allele frequencies of the SNPs were similar to those observed in European populations [57]. The genotype frequencies were in Hardy–Weinberg equilibrium, except for *CD226* rs763361 (*p* < 0.05), which was removed from the analyses. The 1000 Genomes Project Phase 3 linkage disequilibrium data for *CD274* gene variants in European populations are shown in Appendix A.

### 4.5. Immunohistochemistry Analysis

PD-L1 protein expression was assessed through immunohistochemistry on formalin-fixed paraffin-embedded (FFPE) tumor tissue sections using the PD-L1 IHC 22C3 pharmDx kit (Dako Ltd., Cheshire, UK). FFPE tissue sections, 4 µm thick, were deparaffinized and rehydrated. Antigen retrieval was performed by heating the slides for 20 min at 97 °C in a low-pH solution using a PT Link system (Dako Ltd., Cheshire, UK). The sections were then stained automatically with the Dako Autostainer Link 48 (Agilent Technologies, Santa Clara, CA, USA) using the ready-to-use anti-human PD-L1 mouse monoclonal antibody, clone 22C3 (Dako Ltd., Cheshire, UK). This was followed by sequential incubation with a mouse anti-rabbit IgG linker and a secondary antibody and horseradish peroxidase conjugated to a dextran polymer backbone. The slides were then revealed using 3,3′-diaminobenzidine tetrahydrochloride chromogen, counterstained with hematoxylin, and coverslipped. Tonsil tissue served as the positive external control.

Two independent investigators (LL and MF) evaluated the PD-L1 staining, and discordant cases were reviewed for consensus. PD-L1 expression was interpreted using two scoring systems: the TPS and the CPS. The TPS was defined as the percentage of viable tumor cells showing partial or complete membrane staining of any intensity. A specimen was considered PD-L1 positive if the TPS was ≥1%, and high PD-L1 expression was defined as TPS ≥ 50%. The CPS was calculated by dividing the total number of PD-L1-positive cells (membrane-staining tumor cells, cytoplasm-staining lymphocytes, and macrophages) by the total number of viable tumor cells, multiplied by 100. A specimen was considered PD-L1 positive if the CPS was ≥1. Considering the expected low proportion of TPS positivity [35,58], CPS was performed to evaluate PD-L1 expression in tumor-inflammatory immune cells.

### 4.6. Statistical Analysis

Deviations from Hardy–Weinberg equilibrium were assessed for each SNP using the χ^2^ test. Codominant, dominant, or recessive genetic models of inheritance were used for each SNP as appropriate. Survival curves and median survival were calculated using the Kaplan–Meier method. Differences in survival were tested using the log-rank test. Cox proportional hazards regression models were adjusted for multivariate analysis with Eastern Cooperative Oncology Group (ECOG) performance status (PS) (2–4 versus 0–1) and consolidative TRT as covariates. The results are expressed as HRsand 95% CIs. The rate of sensitivity to platinum was analyzed for those SNPs for which we observed significant differences in median PFS in our analyses. Differences in platinum sensitivity were calculated using the χ^2^ test and logistic regression analyses, considering platinum compound (cisplatin or carboplatin) and consolidative TRT as covariates. The results are expressed as ORs with 95% CIs. Progression data for three patients and platinum sensitivity status for four patients were not available and were therefore excluded from the analysis. The size of our sample had a statistical power of 75–87% to detect the effect of polymorphisms with a w = 0.3–0.35. (α = 0.05) (G*power version 3.1.9.7, Düsseldorf, Germany) [59]. Statistical significance was set at *p* ≤ 0.05. Correction for multiple comparisons was set at *p* < 0.001 using the Bonferroni method. Statistical analyses were performed using IBM SPSS Statistics (version 29.0) and the statistical package PLINK (v1.07.2) [60].

## 5. Conclusions

Our results suggest that genetic variants in *CD274* may be predictor biomarkers of survival in ES-SCLC patients treated with platinum- and etoposide-chemotherapy. These results support existing evidence that platinum compounds may be involved in the immunomodulation of the tumor microenvironment. Validation in a larger cohort of patients treated with the current standard treatment of chemotherapy plus IC inhibitors is required.

## Figures and Tables

**Figure 1 ijms-26-04245-f001:**
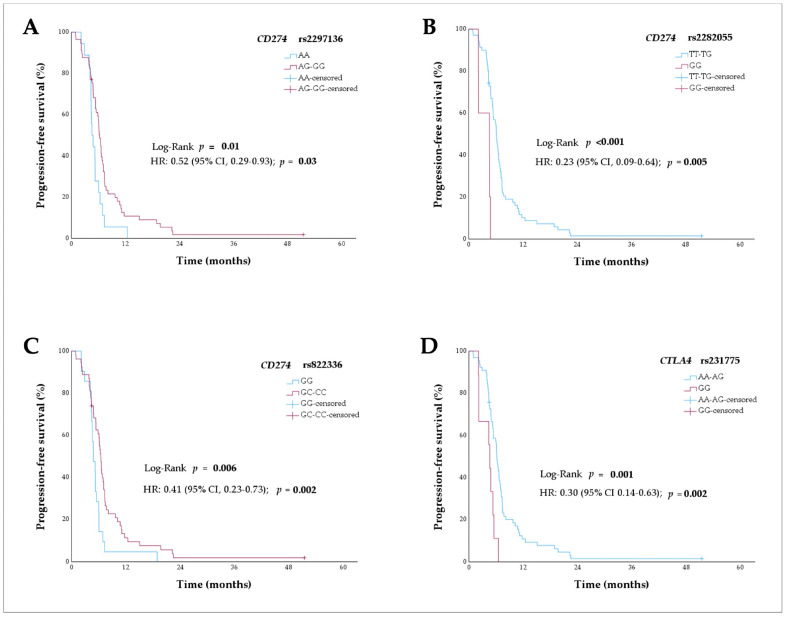
Progression-free survival curves of extensive stage-small cell lung cancer patients (n = 75) according to the (**A**) *CD274* rs2297136 variant in a dominant model of inheritance, (**B**) *CD274* rs2282055 variant in a recessive model of inheritance, (**C**) *CD274* rs822336 variant in a dominant model of inheritance, and (**D**) *CTLA4* rs231775 variant in a recessive model of inheritance.

**Table 1 ijms-26-04245-t001:** Baseline patient characteristics and treatment details.

Characteristic	n = 78
Male (%)	56 (71.8)
Age, median (range), years	65 (38–86)
ECOG performance status (%)	
0	5 (6.4)
1	41 (56.2)
2	22 (28.2)
3–4	8 (10.3)
Unknown	2 (2.6)
Smoking (%)	
Current	52 (66.7)
Former	25 (32.1)
Unknown	1 (1.3)
Brain metastases (%)	16 (21.3)
Liver metastases (%)	35 (44.9)
Chemotherapy regimen selected (%)	
Cisplatin	26 (33.3)
Carboplatin	52 (66.7)
Median number of cycles (range)	6 (2–6)
Consolidative thoracic radiation therapy (%)	19 (24.4)

ECOG, Eastern Cooperative Oncology Group.

**Table 2 ijms-26-04245-t002:** Analyses between the significant immune checkpoint genetic variants and progression-free survival of extensive-stage small cell lung cancer patients.

			Progression-Free Survival			
			Univariate Analysis		Multivariate Analysis	
SNP	n = 75	mPFS (95% CI), Months	HR (95% CI)	*p^a^* Value	HR (95% CI)	*p^b^* Value
*CD274* rs2297136						
AA	18	4.6 (3.8–5.3)	Reference (1)	**0.007**		
AG	43	6.1 (5.2–6.9)	0.58 (0.33–1.02)			
GG	14	6.8 (6.2–7.4)	0.30 (0.14–0.66)			

AA	18	4.6 (3.8–5.3)	Reference (1)			
AG-GG	57	6.3 (5.7–6.8)	0.50 (0.29–0.87)	**0.01** *	0.52 (0.29–0.93)	**0.03**
*CD274* rs2282055						
TT	42	6.1 (5.1–7.0)	Reference (1)	**0.001**		
TG	28	6.1 (4.4–7.8)	0.79 (0.48–1.28)			
GG	5	4.6 (0.0–9.7)	4.48 (1.64–12.23)			

GG	5	4.6 (0.0–9.7)	Reference (1)			
TT-TG	70	6.1 (5.6–6.6)	0.20 (0.08–0.55)	**<0.001** **	0.23 (0.09–0.64)	**0.005**
*CD274* rs822336						
GG	21	4.8 (3.9–5.8)	Reference (1)	**0.02**		
GC	35	6.5 (5.9–7.0)	0.48 (0.27–0.84)			
CC	19	6.6 (5.9–7.3)	0.50 (0.26–0.96)			

GG	21	4.8 (3.9–5.8)	Reference (1)			
GC-CC	54	6.5 (6.0–7.0)	0.48 (0.29–0.82)	**0.006** *	0.41 (0.23–0.73)	**0.002**
*CTLA4* rs231775						
AA	34	6.1 (5.9–6.3)	Reference (1)	**0.005**		
AG	32	6.3 (4.6–7.9)	0.94 (0.58–1.55)			
GG	9	4.6 (3.9–5.3)	3.10 (1.43–6.72)			

GG	9	4.6 (3.9–5.3)	Reference (1)			
AA-AG	66	6.1 (5.7–6.5)	0.31 (0.15–0.07)	**0.001** **	0.30 (0.14–0.63)	**0.002**

SNP, single-nucleotide polymorphism; mPFS, median progression-free survival; HR, hazard ratio; *p^a^* value from a log-rank test; *p^b^* value from a Cox proportional hazards model; * dominant genetic model; ** recessive genetic model. Statistically significant *p*-values are marked in bold.

**Table 3 ijms-26-04245-t003:** Main characteristics of the immune checkpoint genetic variants analyzed.

Gene Symbol	Reference SNP	Variant Description	References for Rationale
*CD274* (*PD-L1*)	rs4143815	3′-UTR	[23,24,34,42]
	rs2297136	3′-UTR	[23,34,43]
	rs2282055	Intron	[42,44]
	rs822336	2 KB Upstream	[24,25,43]
*PDCD1*	rs2227981	Synonymous	[45,46]
	rs10204525	3′-UTR	[47,48]
	rs11568821	Intron	[45,49]
	rs7421861	Intron	[45,48]
*CTLA4*	rs4553808	2 KB Upstream	[50]
	rs231775	Missense	[50,51]
	rs3087243	500B Downstream	[52]
*CD226*	rs763361	Missense	[26,53]
*LAG3*	rs2365095	Intron	[54]
	rs870849	Missense	[55]
	rs3782735	Intron	[56]

SNP, single-nucleotide polymorphism; *CD274*, CD274 molecule; *PD-L1*, programmed cell death ligand 1; *PDCD1*, programmed cell death 1; *CTLA4*, cytotoxic T-lymphocyte associated protein 4; *CD226*, CD226 molecule; *LAG3*, lymphocyte activating 3.

## Data Availability

The data presented in this study include sensitive or confidential information and are not publicly available due to ethical committee regulations but are available upon request from the corresponding authors.

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
