# Peer review of "CD274* (*PD-L1*) Polymorphisms as Predictors of Efficacy in First-Line Platinum-Based Chemotherapy for Extensive-Stage Small Cell Lung Cancer"

_ijms, 2025, doi:10.3390/ijms26094245_

Round 1
Reviewer 1 Report
Comments and Suggestions for Authors
Authors used a retrospective study to examine the effect of CD274 (PD-L1) polymorphisms on among extensive-stage small cell lung cancer patients treated by first-line platinum-based chemotherapy. However, this article has not fully answered some of the questions due to insufficient description.
First, authors suggest “Progression data were not available for three patients.” (L82) and “Platinum sensitivity status was not available for four patients.” (L93), but they do not explain how to use them in their analyses (e.g., they were excluded from their analyses). Without explanation, it is difficult to understand what authors did. Authors should add the explanation in method section.
Second, authors added p-values (i.e., 0.007, 0001, 0.02, and 0.005) next to references in Table 2, but it is difficult to understand what they were. Authors should add explanation for these P-values.
Finally, authors described some of sentences without citation or justification as follows; “Platinum-based chemotherapy plus etoposide is the cornerstone of treatment for patients with untreated small cell lung carcinoma (SCLC).” (L34), “ES-SCLC is an aggressive disease with a poor prognosis.” (L166), “The CD274 gene encodes the immune inhibitory ligand PD-L1 which binds to the PD-1 receptor blocking T-cell activation to prevent autoimmunity, a mechanism also used by the tumor to escape immune surveillance.” (L194), “CD274 rs2282055 is an intron variant.” (L), “The CTLA4 rs231775 A>G is a missense polymorphism (p.Thr17Ala) located in exon 1.” (L223), “Platinum compounds have a cytotoxic effect on cancer cells but also act as host immune modulators.” (L228), and “RECIST 1.1” (L258), but it is difficult for readers to judge them without references as evidence for each description. Authors should add references for these descriptions.
Minor comments
Table 1: Unit for “Cycles median” (e.g., days) should be added.
Reviewer 2 Report
Comments and Suggestions for Authors
In this study the authors address PDL1 as predictive marker for effective lung cancer therapy. Despite , the manuscript is well written, some points should be clarified before publishing.
Title: please delete PD-L1 from the title.
Abstract, please rephrase and minimize the numerical values.
Introduction, please add information about lung cancer risk factors, overview of predictive genetic and proteomics biomarkers anticancer monitoring and mechanism of action platinum compounds and etoposide as anticancer agents. Please add the study rationale.
Results, please include the statical data in the tables and figures legends ( samples number, data expression, significance, and p value).
Discussion, please refine this section and support your work with similar finding and add the contradictory if present. Please discuss the study limitations.
Methods, please add details about inclusion and exclusion criteria , therapeutic protocol and ethical approval of the study .
Please add reference 2025 citation dated.
Please , check the manuscript for misuse of acronyms as well as long sentences or paragraphs without references citation.
Please, minimize the use of personal pronouns in your writing.
Comments on the Quality of English LanguagePlease, check the manuscript for minor grammar errors and syntax.
Round 2
Reviewer 1 Report
Comments and Suggestions for Authors
Authors revised the manuscript, but this article has not fully answered some of the questions due to insufficient description.
In fact, as mentioned in the previous review, authors added p-values (i.e., 0.007, 0001, 0.02, and 0.005) next to references in Table 2, but it is difficult to understand what they were. Authors added footnote, but it is difficult to understand the differences between these p-values and other p-values (i.e., 0.01, < 0.001, 0.006, and 0.001). Authors should add explanation for these P-values.
Round 3
Reviewer 1 Report
Comments and Suggestions for Authors
I have no further comments.